# Analysis of Influence of Behavioral Adoption Threshold Diversity on Multi-Layer Network

**DOI:** 10.3390/e25030458

**Published:** 2023-03-06

**Authors:** Gang Deng, Yuting Peng, Yang Tian, Xuzhen Zhu

**Affiliations:** 1School of Information and Communication Engineering, Beijing University of Posts and Telecommunications, Beijing 100876, China; 2State Key Laboratory of Networking and Switching Technology, Beijing University of Posts and Telecommunications, Beijing 100876, China

**Keywords:** complex networks, information propagation, multi-layer networks, adoption behavior diversity

## Abstract

The same people exhibit various adoption behaviors for the same information on various networks. Previous studies, however, did not examine the variety of adoption behaviors on multi-layer networks or take into consideration this phenomenon. Therefore, we refer to this phenomenon, which lacks systematic analysis and investigation, as behavioral adoption diversity on multi-layered networks. Meanwhile, individual adoption behaviors have LTI (local trend imitation) characteristics that help spread information. In order to study the diverse LTI behaviors on information propagation, a two-layer network model is presented. Following that, we provide two adoption threshold functions to describe diverse LTI behaviors. The crossover phenomena in the phase transition is shown to exist through theoretical derivation and experimental simulation. Specifically, the final spreading scale displays a second-order continuous phase transition when individuals exhibit active LTI behaviors, and, when individuals behave negatively, a first-order discontinuous phase transition can be noticed in the final spreading scale. Additionally, the propagation phenomena might be impacted by the degree distribution heterogeneity. Finally, there is a good agreement between the outcomes of our theoretical analysis and simulation.

## 1. Introduction

The modern world has entered the Internet era, and social networks such as QQ, WeChat, Ins, and Douyin have grown in popularity and become vital tools in people’s businesses and personal lives [1,2,3]. In social networks, various kinds of information are propagated among users, so that people can obtain new information in time [4,5]. More and more academics are studying the mechanism of information propagation in order to provide more practical application value [6,7,8]. At present, information propagation has been used in the domains of aerospace [9], transportation [10], agriculture [11], medicine [12], and other fields.

After extensive research on the mechanism of information propagation, it has been discovered that certain factors, including behavior adoption threshold [13], degree heterogeneity [14], node distribution structure [15], etc., will affect information propagation. This finding has been supported by both theoretical and experimental evidence. Specifically, Wang et al. proposed a social contagion model with adoption probability thresholds that look similar to gates and consist of “on” and “off” [16]. Then, Zhu et al. presented a new model based on a two-layer network, where information is synchronously conveyed on both layers and each layer is provided with a distinct adoption threshold, to investigate the effect of population heterogeneity on social contagion [17]. Furthermore, an information propagation model with limited contact is proposed in Yu’s paper [18]. A two-layer weighted social network model was also provided by Zhu et al. in 2019, and it is based on the threshold model [19]. Then, Zhu et al. proposed a two-layer network model with edge weight distribution to explore the effect of the heterogeneous adoption threshold of individuals on information propagation [20].

Information propagation is very closely linked to physical networks. Individual interactions in the social world and many other scenarios can be represented by multi-layer networks [21]. The study of multi-layer networks can also reveal some physical phenomena. The relationships between family members, close friends, and work associates form the structure of multi-layer networks. For example, the connectivity of different layers of multi-layer networks is also reflected in natural and human-made transportation systems [22]. At the same time, the process of information propagation also has a reinforcement effect because it necessitates repeated confirmation to guarantee the accuracy and reliability of the information [23]. In order to analyze the reinforcement effect in social networks, scholars have proposed many classical models, including Markovian and non-Markovian propagation models [24]. The threshold model, as a model of the classical non-Markov process, is often used to describe the reinforcement effect of information propagation [25]. On the basis of the threshold model, scholars have proposed other threshold models that conform to real networks, and verified them through experiments [18,20]. On the other side, after the information has been effectively transmitted, an edge in the network will stop transmitting further information [26,27].

It has been found that different social networks exhibit a variety of structural traits, including edge weight heterogeneity, individual contact heterogeneity, degree heterogeneity, and propagation probability heterogeneity. The same person exhibits various adoption intentions and actions on various network tiers as a result of the various social network attributes. An event might simultaneously spread via Weibo and WeChat, for instance. WeChat is an instant messaging app that supports voice, video, picture, and text transmission, and is more often shared between friends and loved ones. Weibo is a social media platform based on user relationships, which can realize the sharing and interaction of information, so that users are closely connected with the world, and the information spreads faster and the audience is wider. Weibo’s huge viewership helps the incident spread more quickly. The event, however, demonstrates greater authenticity because Wechat has a higher level of intimacy. Thus, both different networks have an impact on an individual’s activity. In the real world, the same individuals exhibit limited social skills, while, on the virtual network layer, they exhibit strong social conduct. The above examples illustrate that information propagation on multi-layer networks is characterized by behavioral diversity. Individuals display LTI behaviors in a social contagion model built on a single-layer network, claimed a study by Zhu et al. [28]. However, their research did not consider multi-layer networks, it had an insufficient grasp of the actual scene changes, and it only explored the information propagation on a single-layer network. In fact, the same individuals on different networks show different behaviors, so it is necessary to establish different threshold models on the two-layer networks to fit the actual scene better. Based on the active and negative behaviors of the same individuals on different networks, we define the multi-layer networks as the A-layer and B-layer. When individuals are highly active on the A-layer network, they exhibit active LTI behavior across a variety of networks. On the B-layer network, however, when individuals have low activity, they exhibit a negative LTI behavior. Therefore, we explore and analyze the impact of adoption threshold diversity on information propagation on multi-layer networks. Through simulations, we verify that, on multi-layer networks, the multiple LTI behaviors affect the propagation process.

The remainder of the essay is structured as follows: In Section 2, we construct a propagation model with multiple adoption thresholds based on multi-layer networks. Section 3 shows an improved theoretical analysis. Section 4 shows the related parameters. In Section 5, we discuss the simulations and experimental results. Section 6 exhibits the conclusion and description.

## 2. Multi-Layer Network Models

We build a two-layer network model with diverse adoption thresholds and LTI characteristics to examine the effects of adoption behavior diversity on various networks. The A-layer and B-layer networks are two separate networks, which are coupled to each other by the same individual, and the individual possesses different adoption thresholds on different layers. The scholars proposed a generalized Susceptible-Adopted-Recovered (SAR) [29,30] model to represent the information propagation mechanism in the two-layer networks, as illustrated in Figure 1a. In the SAR model, each node in the network has three different states, namely, the susceptible state, adopted state, and recovered state, corresponding to the S-state, A-state, and R-state, respectively. The S-state means that an individual has the tendency to adopt behavior, and can accept information from surrounding nodes, but has not yet embraced the information behavior. The A-state indicates that an individual adopts the information behavior and is eager to share information with its nearby neighbors. The R-state indicates that the individuals with the A-state are no longer interested in the information. The individuals exhibit different adoption behaviors on different layers, which affect the information propagation mechanism.

We present two distinct threshold functions with LTI features to illustrate the impact of behavioral diversity on information propagation on the two-layer network, as illustrated in Figure 1b. Everyone has different levels of receptivity to new things, but if something is popular at the moment and someone’s friends in their circle start mentioning and recommending it, then individuals will gradually increase their level of knowledge about it and increase the likelihood of adopting it until it is fully adopted. It is as if the individual is in an active network similar to the A-layer network, starting out at a very low level and eventually reaching its maximum value. The threshold model can be expressed by
(1)$$h_{A}\left(x\right)=\left\{\begin{array}{c} \frac{x}{a},\;\;\;0<x\leq a\\ 1,\;\;\;a<x\leq 1 \end{array}\right.$$
where $h_{A}\left(x\right)$ is the proportion of the total number of information $m_{A}$ received by the S-state individual to its degree in the A-layer, and *a* denotes the individual’s LTI parameter in the A-layer.

The individuals maintain a neutral attitude toward a certain message on Weibo, and the degree of reception gradually decreases to 0 due to the bias brought by Weibo marketing numbers, which leads to the aversion of individuals to this message. It is as if, in the B-layer network, individuals are not active or even resistant to adopting information, so they have never adopted behavior. When the number of transmissions is greater than the adoption threshold, the individuals will adopt the behavior. However, because they are resistant, they will refuse to transmit information to the surrounding neighbors. The adoption scale will also eventually drop to 0.

The threshold model can be expressed as
(2)$$h_{B}\left(x\right)=\left\{\begin{array}{c} 0,\;\;\;0<x\leq b\\ \frac{1-x}{1-b},\;b<x\leq 1 \end{array}\right.,$$
where $h_{B}\left(x\right)$ is the proportion of the total number of information $m_{B}$ received by the S-state individual to its degree in the B-layer, and *b* denotes the individual’s LTI parameter in the B-layer.

Although the structures of the two layers of the network are different, there is no conflict. That is, the individuals do not need to choose only one of them for information propagation, similar to how individuals can retweet and comment on both Weibo and WeChat, and, thus, are also positively influenced by the A-layer network and negatively influenced by the B-layer network.

In reality, as people are on different networks, the adoption behavior of the same information shows diverse characteristics. When the propagation scenario changes, they will refresh their perception of the information and change their adoption behavior for the information. The possibility that they will adopt it rises as they acquire more information at each layer of the network. The process of information propagation in the two-layer networks can be summed up in the following way: we suppose that, in the multi-layer networks, ρ individuals are set as the initial infection seeds in the A-state, while others are in the S-state. The individuals in the A-state transfer information to their neighbors in the S-state by the edge transmission probability λ. Once individuals in the S-state receive information, the number of receptions will accumulate until the individual changes to A-state. On one layer of the network, when an individual changes their state, the other layer of the network also changes. As information propagation is non-redundant, the information cannot be successfully sent twice on the same edge. After the individual with the A-state successfully transmits the information, it will convert with probability γ to the R-state. When no individual on the two-layer networks is in the A-state, the information propagation process is complete.

## 3. Theoretical Analysis

On the multi-layer networks, the individuals who link the diverse layered networks are connected one to one, so the same individual is present in each of these layers. Thus, $\vec{k_{i}}=\left(k_{i}^{A},k_{i}^{B}\right)$ represents the degree vector of individual *i*, where $k_{i}^{X}$(X∈{A,B}) represents the node degree of the A-layer or B-layer. $P(\vec{k})$ with the degree vector $\vec{k}$ represents the degree distribution. Additionally, the different network layers also contain a large number of separate connections. Consequently, by the uncorrelated characteristic, $P(\vec{k})$ in layer *A* or *B* can decompose into degree distributions $P_{X}\left(k^{X}\right)$. As a result of the independence of $P_{A}\left(k^{A}\right)$ and $P_{B}\left(k^{B}\right)$, $P\left(\vec{k}\right)=P_{A}\left(k^{A}\right)P_{B}\left(k^{B}\right)$.

We look into how the adoption threshold diversity affects the way information is spread throughout the multi-layer networks and suggests an enhanced edge partition theory to analyze the information propagation mechanism. It is assumed that there is a node in the network that is in a cavity state [31], which implies that it can receive information from its neighbors but cannot transmit information to the outside world. In this paper, *i* represents a random individual, and *j* denotes the neighbors of *i*. $\theta_{k_{j}^{X}}^{X}\left(t\right)$ represents the probability that *j* randomly selects an edge in *X*-layer that does not transmit information to *i* at *t*, then 1-$\theta_{k_{j}^{X}}^{X}\left(t\right)$ represents the probability that *j* transmits information to *i* at this time. The probability that *i* and *j* with a degree of $k_{j}^{X}$ are neighbors in *X*-layer is indicated by $\frac{k_{j}^{X}P\left(k_{j}^{X}\right)}{\langle k^{X}\rangle }$. Thus, individual *i* cannot obtain the information from *j* of the *X*-layer by time *t* with the probability
(3)$$\theta_{X}\left(t\right)=\sum_{k_{j}^{X}=0}\frac{k_{j}^{X}P\left(k_{j}^{X}\right)}{\langle k^{X}\rangle }\theta_{k_{j}^{X}}^{X}\left(t\right).$$

Thus, we can obtain the probability that the individual *i* with degree $\vec{k}=\left(k_{i}^{A},k_{i}^{B}\right)$ receives $m_{X}$ units of information from the surrounding neighbors *j* in the *X*-layer is the following equation:(4)$$\phi_{m_{X}}^{X}\left(k_{i}^{X},t\right)=\left(\begin{array}{c} k_{i}^{X}\\ m_{X} \end{array}\right)\theta_{X}{\left(t\right)}^{k_{i}^{A}-m_{X}}{\left[1-\theta_{X}\left(t\right)\right]}^{m_{X}},$$

At the same time, these individuals also maintain the S-state with the probability of $\overset{m_{A}}{\underset{j=0}{\Pi }}\left[1-h_{A}\left(\frac{j}{k_{i}^{A}},a\right)\right]$ in layer *A* and $\overset{m_{B}}{\underset{j=0}{\Pi }}\left[1-h_{B}\left(\frac{j}{k_{i}^{B}},b\right)\right]$ in layer *B*. The probability that an individual will still be in the S-state at time *t* after accumulative accepting $m_{A}$ and $m_{B}$ pieces of information from those of layer *A* and *B* can be represented as
(5)$$\begin{array}{c} \begin{array}{c} \varsigma_{m_{A}}^{A}\left(k_{i}^{A},t\right)=\sum_{m_{A}=0}^{k_{i}^{A}}\phi_{m_{A}}^{A}\left(k_{i}^{A},t\right)\prod_{j=0}^{m_{A}}\left[1-h_{A}\left(\frac{j}{k_{i}^{A}},a\right)\right]\\ \;\;\;\;=\sum_{m_{A}=0}^{\left[ak_{i}^{A}\right]}\phi_{m_{A}}^{A}\left(k_{i}^{A},t\right)\prod_{j=0}^{m_{A}}\left[1-\frac{j}{ak_{i}^{A}}\right]\\ \;\;\;\;+\sum_{m_{A}=\left[ak_{i}^{A}\right]}^{k_{i}^{A}}\phi_{m_{A}}^{A}\left(k_{i}^{A},t\right)\prod_{j=0}^{\left[ak_{i}^{A}\right]}\left[1-\frac{j}{ak_{i}^{A}}\right]\prod_{j=[ak_{i}^{A}]}^{m_{A}}\left(1-1\right) \end{array}\\ \;\;\;\;\;=\sum_{m_{A}=0}^{\left[ak_{i}^{A}\right]}\phi_{m_{A}}^{A}\left(k_{i}^{A},t\right)\prod_{j=0}^{m_{A}}\left[1-\frac{j}{ak_{i}^{A}}\right] \end{array}$$
and
(6)$$\begin{array}{c} \begin{array}{c} \varsigma_{m_{B}}^{B}\left(k_{i}^{B},t\right)=\sum_{m_{B}=0}^{k_{i}^{B}}\phi_{m_{B}}^{B}\left(k_{i}^{B},t\right)\prod_{j=0}^{m_{B}}\left[1-h_{B}\left(\frac{j}{k_{i}^{B}},b\right)\right]\\ \;\;\;\;=\sum_{m_{B}=0}^{\left\lfloor bk_{i}^{B}\right\rfloor }\phi_{m_{B}}^{B}\left(k_{i}^{B},t\right)\prod_{j=0}^{m_{B}}\left[1-0\right]\\ \;\;\;\;+\sum_{m_{B}=\left\lceil bk_{i}^{B}\right\rceil }^{k_{i}^{B}}\phi_{m_{B}}^{B}\left(k_{i}^{B},t\right)\prod_{j=0}^{\left\lceil bk_{i}^{B}\right\rceil }\left[1-0\right]\prod_{j=\left\lceil bk_{i}^{B}\right\rceil }^{m_{B}}\left(1-\frac{1-\frac{j}{k_{i}^{B}}}{1-b}\right) \end{array}\\ \;\;\;\;=\sum_{m_{B}=0}^{\left\lfloor bk_{i}^{B}\right\rfloor }\phi_{m_{B}}^{B}\left(k_{i}^{B},t\right)+\sum_{m_{B}=\left\lceil bk_{i}^{B}\right\rceil }^{k_{i}^{B}}\phi_{m_{B}}^{B}\left(k_{i}^{B},t\right)\prod_{j=\left\lceil bk_{i}^{B}\right\rceil }^{m_{B}}\left(1-\frac{1-\frac{j}{k_{i}^{B}}}{1-b}\right) \end{array},$$
respectively.

Therefore, when individual *i* receives $m_{A}$ and $m_{B}$ units of information, the probability that they remain S-state is:(7)$$\begin{array}{c} s\left(\vec{k},t\right)=\left(1-\rho_{0}\right)\sum_{m_{A}=0}^{k_{i}^{A}}\phi_{m_{A}}^{A}\left(k_{i}^{A},t\right)\overset{m_{A}}{\underset{j=0}{\Pi }}\left[1-h_{A}\left(\frac{j}{k_{i}^{A}},a\right)\right]\\ \;\;\;\;\times \sum_{m_{B}=0}^{k_{i}^{B}}\phi_{m_{B}}^{B}\left(k_{i}^{B},t\right)\overset{m_{B}}{\underset{j=0}{\Pi }}\left[1-h_{B}\left(\frac{j}{k_{i}^{B}},b\right)\right]\\ \;\;\;\;=\left(1-\rho_{0}\right)\varsigma_{m_{A}}^{A}\left(k_{i}^{A},t\right)\varsigma_{m_{B}}^{B}\left(k_{i}^{B},t\right), \end{array}$$

When the information has not been accepted by the S-state individuals, the possibility of the accumulated information in the A-layer or B-layer by time *t* is denoted as
(8)$$\eta_{X}=\sum_{k_{i}^{X}}P_{X}\left(k_{i}^{X}\right)\varsigma_{m_{X}}^{X}\left(k_{i}^{X},t\right),$$

Therefore, the proportion of the individuals in the multi-layer networks, which are in the S-state, at time *t* is denoted as
(9)$$S\left(t\right)=\sum_{\vec{k}}P\left(\vec{k}\right)s\left(\vec{k},t\right)=\left(1-\rho_{0}\right)\eta_{A}\eta_{B},$$

Furthermore, because the individuals may be in the S-state, A-state, or R-state, $\xi_{S,k_{j}^{X}}^{X}\left(t\right)$, $\xi_{A,k_{j}^{X}}^{X}\left(t\right)$, and $\xi_{R,k_{j}^{X}}^{X}\left(t\right)$ represent the probability of being in the S-state, A-state, and R-state, respectively, so $\theta_{k_{j}^{X}}^{X}\left(t\right)$ can be decomposed into:(10)$$\theta_{k_{j}^{X}}^{X}\left(t\right)=\xi_{S,k_{j}^{X}}^{X}\left(t\right)+\xi_{A,k_{j}^{X}}^{X}\left(t\right)+\xi_{R,k_{j}^{X}}^{X}\left(t\right),$$

However, because of the cavity theory, the individual *i* in the cavity state is unable to communicate with its neighbors. The neighbor *j*’s degree vector is $\vec{k_{j}}=\left(k_{j}^{A},k_{j}^{B}\right)$. If the S-state individual *i* connects its neighbor *j* in the *A*-layer, then all $k_{j}^{A}-1$ neighbors, except individual *i* in the *A*-layer and $k_{j}^{B}$ neighbors in the *B*-layer, can share information. $\tau_{n_{A}}^{A}\left(k_{j}^{A}-1,t\right)$ and $\tau_{n_{B}}^{B}\left(k_{j}^{B}-1,t\right)$ represent the probability that the neighbor *j* with degree $\vec{k_{j}}=\left(k_{j}^{A},k_{j}^{B}\right)$ has received $n_{A}$ bits of information from its neighbors up to time *t*. The probability can be calculated by
(11)$$\begin{array}{c} \begin{array}{c} \tau_{n_{A}}^{A}\left(k_{j}^{A}-1,t\right)=\sum_{n_{A}=0}^{k_{i}^{A}-1}\phi_{n_{A}}^{A}\left(k_{j}^{A}-1,t\right)\prod_{j=0}^{n_{A}}\left[1-h_{A}\left(\frac{j}{k_{j}^{A}-1},a\right)\right]\\ =\sum_{n_{A}=0}^{\left\lfloor ak_{j}^{A}\right\rfloor }\phi_{n_{A}}^{A}\left(k_{j}^{A}-1,t\right)\prod_{j=0}^{n_{A}}\left[1-\frac{j}{ak_{j}^{A}}\right]\\ +\sum_{n_{A}=\left\lceil ak_{j}^{A}\right\rceil }^{k_{i}^{A}-1}\phi_{n_{A}}^{A}\left(k_{j}^{A}-1,t\right)\prod_{j=0}^{\left\lceil ak_{j}^{A}\right\rceil }\left[1-\frac{j}{ak_{j}^{A}}\right]\prod_{j=\left\lceil ak_{j}^{A}\right\rceil }^{n_{A}}\left(1-1\right) \end{array}\\ =\sum_{n_{A}=0}^{\left\lfloor ak_{j}^{A}\right\rfloor }\phi_{n_{A}}^{A}\left(k_{j}^{A}-1,t\right)\prod_{j=0}^{n_{A}}\left[1-\frac{j}{ak_{j}^{A}}\right] \end{array}$$
and
(12)$$\begin{array}{c} \begin{array}{c} \tau_{n_{B}}^{B}\left(k_{j}^{B}-1,t\right)=\sum_{n_{B}=0}^{k_{j}^{B}-1}\phi_{n_{B}}^{B}\left(k_{j}^{B}-1,t\right)\prod_{j=0}^{n_{B}}\left[1-h_{B}\left(\frac{j}{k_{i}^{A}-1},b\right)\right]\\ =\sum_{n_{B}=0}^{\left\lfloor bk_{j}^{B}\right\rfloor }\phi_{n_{B}}^{B}\left(k_{j}^{B}-1,t\right)\prod_{j=0}^{n_{B}}\left[1-0\right]\\ +\sum_{n_{B}=\left\lceil bk_{j}^{B}\right\rceil }^{k_{j}^{B}}\phi_{n_{B}}^{B}\left(k_{j}^{B}-1,t\right)\prod_{j=0}^{\left\lceil bk_{j}^{B}\right\rceil }\left[1-0\right]\prod_{j=\left\lceil bk_{j}^{B}\right\rceil }^{m_{B}}\left(1-\frac{1-\frac{j}{k_{j}^{B}}}{1-b}\right) \end{array}\\ =\sum_{n_{B}=0}^{\left\lfloor bk_{j}^{B}\right\rfloor }\phi_{n_{B}}^{B}\left(k_{j}^{B}-1,t\right)+\sum_{n_{B}=\left\lceil bk_{j}^{B}\right\rceil }^{k_{j}^{B}}\phi_{n_{B}}^{B}\left(k_{j}^{B}-1,t\right)\prod_{j=\left\lceil bk_{j}^{B}\right\rceil }^{n_{B}}\left(1-\frac{1-\frac{j}{k_{j}^{B}}}{1-b}\right), \end{array}$$
respectively.

Additionally, the likelihood that an individual *j* with degree $\vec{k_{j}}=\left(k_{j}^{A},k_{j}^{B}\right)$ has acquired $n_{B}$ bits of information from neighbors of *B*-layer by the time *t* is $\varsigma_{n_{B}}^{B}\left(k_{j}^{B},t\right)$. Therefore, the probability that the individual *j* will remain in the S-state after receiving $n_{A}$ and $n_{B}$ pieces of information cumulatively is
(13)$$\begin{array}{c} \theta_{A}\left(\vec{k},t\right)=\sum_{n_{A}=0}^{k_{j}^{A}-1}\phi_{n_{A}}^{A}\left(k_{j}^{A}-1,t\right)\overset{n_{A}}{\underset{j=0}{\Pi }}\left[1-h_{A}\left(\frac{j}{k_{j}^{A}},a\right)\right]\\ \;\;\;\;\times \sum_{n_{B}=0}^{k_{j}^{B}}\phi_{n_{B}}^{B}\left(k_{j}^{B},t\right)\overset{n_{B}}{\underset{j=0}{\Pi }}\left[1-h_{B}\left(\frac{j}{k_{j}^{B}},b\right)\right]\\ \;\;\;\;=\tau_{n_{A}}^{A}\left(k_{j}^{A}-1,t\right)\varsigma_{n_{B}}^{B}\left(k_{j}^{B},t\right) \end{array}$$

When the individual *i* in the S-state interacts with its neighbor *j* by a degree of $\vec{k_{j}}$, the probability that the individual *j* in the *B*-layer will remain in the S-state after obtaining all of the $n_{A}$ and $n_{B}$ pieces of information is
(14)$$\begin{array}{c} \theta_{B}\left(\vec{k},t\right)=\sum_{n_{A}=0}^{k_{j}^{A}}\phi_{n_{A}}^{A}\left(k_{j}^{A},t\right)\overset{n_{A}}{\underset{j=0}{\Pi }}\left[1-h_{A}\left(\frac{j}{k_{j}^{A}},a\right)\right]\\ \;\;\;\;\times \sum_{n_{B}=0}^{k_{j}^{B}-1}\phi_{n_{B}}^{B}\left(k_{j}^{B}-1,t\right)\overset{n_{B}}{\underset{j=0}{\Pi }}\left[1-h_{B}\left(\frac{j}{k_{j}^{B}},b\right)\right]\\ \;\;\;\;=\tau_{n_{A}}^{A}\left(k_{j}^{A},t\right)\varsigma_{n_{B}}^{B}\left(k_{j}^{B}-1,t\right) \end{array}$$

The network is uncorrelated, so the probability that an edge is connected to an individual of degree $k_{j}^{X}$ is $k_{j}^{X}P_{X}\left(\vec{k_{j}}\right)/\langle k^{X}\rangle$, while $\langle k\rangle$ is the average degree of the network; then, we can obtain the probability of being in the S-state:(15)$$\xi_{S,k_{j}^{X}}^{X}\left(t\right)=\left(1-\rho_{0}\right)\sum_{\vec{k_{j}}}\frac{k_{j}^{X}P_{X}\left(\vec{k_{j}}\right)}{\langle k^{X}\rangle }\theta_{X}\left(\vec{k_{j}},t\right)$$

If the A-state transmits behavioral information with the probability of λ, then $\theta_{k_{j}^{X}}^{X}\left(t\right)$ will decrease by a proportion equal to $\lambda \xi_{A,k_{j}^{X}}^{X}\left(t\right)$, namely:(16)$$\frac{d\theta_{k_{j}^{X}}^{X}\left(t\right)}{dt}=-\lambda \xi_{A,k_{j}^{X}}^{X}\left(t\right)$$

The individuals in the A-state stop caring about the spreading information and become R-state in a possibility of γ. The calculation for $\xi_{R,k_{j}^{X}}^{X}\left(t\right)$’s evolution is
(17)$$\frac{d\xi_{R,k_{j}^{X}}^{X}\left(t\right)}{dt}=\gamma \left(1-\lambda \right)\xi_{A,k_{j}^{X}}^{X}\left(t\right)$$

According to the initial conditions $\theta_{k_{j}^{X}}^{X}\left(0\right)=1$, $\xi_{R,k_{j}^{X}}^{X}\left(0\right)=0$, we can obtain:(18)$$\xi_{R,k_{j}^{X}}^{X}\left(t\right)=\frac{\gamma \left[1-\theta_{k_{j}^{X}}^{X}\left(t\right)\right]\left(1-\lambda \right)}{\lambda }$$

Substitute $\xi_{S,k_{j}^{X}}^{X}\left(t\right)$, $\xi_{A,k_{j}^{X}}^{X}\left(t\right)$, $\xi_{R,k_{j}^{X}}^{X}\left(t\right)$, respectively, to obtain the time change in $\theta_{k_{j}^{X}}^{X}\left(t\right)$:(19)$$\frac{d\theta_{k_{j}^{X}}^{X}\left(t\right)}{dt}=-\lambda \left[\theta_{k_{j}^{X}}^{X}\left(t\right)-\sum_{\vec{k_{j}}}\frac{k_{j}^{X}P_{X}\left(\vec{k_{j}}\right)}{\langle k^{X}\rangle }\theta_{X}\left(\vec{k_{j}},t\right)\right]+\gamma \left[1-\theta_{k_{j}^{X}}^{X}\left(t\right)\right]\left(1-\lambda \right)$$

At the moment of transmitting the behavioral information, some susceptible individuals choose to adopt the behavior, and some adopted individuals will enter the R-state. Thus, the time change in A(t) is:(20)$$\frac{dA\left(t\right)}{dt}=-\frac{dS(t)}{dt}-\gamma A\left(t\right)$$
(21)$$\frac{dR(t)}{dt}=\gamma A\left(t\right)$$

The social communication phenomenon can be described by the above formula, and the proportion of the S-state, A-state, and R-state can be obtained at any time. When time t approaches infinity, the final adoption scale R(∞) can be obtained:(22)$$\theta_{k_{j}^{X}}^{X}\left(\infty \right)=\sum_{\vec{k_{j}}}\frac{k_{j}^{X}P_{X}\left(\vec{k_{j}}\right)}{\langle k^{X}\rangle }\theta_{X}\left(\vec{k_{j}},\infty \right)+\frac{\gamma \left[1-\theta_{k_{j}^{X}}^{X}\left(\infty \right)\right]\left(1-\lambda \right)}{\lambda }$$

It can be seen from the $\theta_{k_{j}^{X}}^{X}\left(t\right)$ equation that, when the A-state continuously transmits information to the surrounding neighbors, $\theta_{k_{j}^{X}}^{X}\left(t\right)$ will decrease with time. When $\theta_{k_{j}^{X}}^{X}\left(t\right)$ is at the maximum stable point, substitute this value to obtain the stable value of the susceptible state S(∞) and the final adoption scale R(∞).

## 4. Related Parameters

In this section, tests on the multi-layer ER network [32] and SF network [33] are used to simulate and assess the proposed model. The ER(ErdOs-Renyi) random network is an equal opportunity network model, i.e., given a certain number of nodes, the probability of inter connection with other surrounding nodes is the same. Since the connection probability of a single node with *k* other nodes decreases exponentially as the value of *k* increases, the connection probability is subject to Poisson distribution. The SF network refers to the scale-free network. Most of the networks in the real world are not random networks, where a few nodes have a large number of connections while most of them have few connections, so the degree distribution of nodes conforms to a power–law distribution, which is also called scale-free property. Complex networks with degree distribution conforming to power–law distribution become scale-free networks. We set 10,000 nodes on the ER network and the SF network as propagation individuals. Furthermore, each network has an average degree of $\langle k_{A}\rangle =\langle k_{B}\rangle =10$. The probability of unit information spreading is expressed at λ. In the ER network, the probability of connecting any two nodes is the same, and the degree of the nodes in the *X*-layer obeys the Poisson distribution $P_{X}\left(k_{X}\right)=e^{-\langle k_{X}\rangle }\frac{{\langle k\rangle }^{k_{X}}}{k_{X}!}$. In the SF network, the degree exponent *v* has a negative correlation with the heterogeneity of the nodes’ degree distributions, and where $\zeta_{X}=\frac{1}{\sum_{k_{X}}{k_{X}}^{-v}}$, the degrees of nodes follow a power–law distribution with $P_{X}\left(k_{X}\right)=\zeta_{X}{k_{X}}^{-v}$. In the experiment, we first set ten initial A-state (initial seed ratio is $h_{0}=0.001$) individuals as the source of transmission. The individuals in the A-state have a probability of γ = 1.0 of returning to the R-state.

Furthermore, the relative variance is unitized, which is written as follows, to demonstrate the crucial condition in our simulation.
(23)$$\chi =N\frac{\langle R{\left(\infty \right)}^{2}\rangle -{\langle R(\infty )\rangle }^{2}}{\langle R(\infty )\rangle },$$
where 〈…〉 stands for the mean set. The important points of the final adoption scale are implied by the χ peak values.

## 5. Experiments and Discussions

### 5.1. Propagation Phenomena on a Two-Layer ER Network

Figure 2 shows the time evolution diagram of the individuals in the S-state, A-state, and R-state. The parameters of (a) and (b) are adopted as a=b=0.1 and a=b=0.3, respectively. In general, the network starts off with only S-state individuals and steadily decreases to 0 over time. The proportion of individuals in the A-state gradually increase to the highest peak, and decrease to 0. Over time, the R-state steadily moves up to the final adoption scale. As the adoption threshold parameter increases from a=b=0.1 in subgraph (Figure 2a) to a=b=0.3 in subgraph (Figure 2b), the cost of the evolution time continuously rises from 6 to 8. The phenomena illustrate that information spreading on a multi-layer network can be promoted by individual behavior reinforcement (active adoption behavior).

Figure 3 shows the effects of unit transmission probability along with two behavioral parameters *a* and *b* on each individual’s final adoption scale in the multi-layer ER network. We can see from Figure 3c that, as λ increases, the final spreading scale R(∞) increases to global spreading. Furthermore, Figure 3c also indicates how the diverse behaviors have an effect on the phase transition. When an individual shows an active behavior on the multi-layer networks simultaneously, e.g., a=b=0.2, the pattern of R(∞) reveals a second-order growth in the continuous phase transition, suggesting that, even though small λ, a strong behavior can lead to global propagation. When a=b=0.5 and a=b=0.8, however, R(∞) reveals a first-order growth in the discontinuous phase transition while an individual shows a negative behavior on the multi-layer networks. This suggests that the individual with a weak adoption behavior slows the spread of information.

To further observe the critical phenomena around the phase transition, we reduce the network_size to 1000 and add the network_size to 15,000 to obtain Figure 3a,b and Figure 3e,f. We find from Figure 3a that global adoption can be achieved with smaller λ when the network_size is reduced. Furthermore, again at a=b=0.2, a second-order continuous phase transition is revealed, and, at a=b=0.5 and a=b=0.8, a first-order discontinuous phase transition is produced. On the contrary, when the network_size is increased to 15,000, as shown in Figure 3e, the global adoption can be achieved with larger λ, and the type of phase transition produced is similar to the former. Therefore, as the network_size increases, the λ that generates the global adoption also increases, transitioning from a second-order continuous phase transition to a first-order discontinuous phase transition.

The relative variances and crucial information spreading possibilities of (a,c,e), are shown in Figure 3b,d,f. The deviation of information spreading, which is depicted by the top values of relative variance χ, is where the global adoption information will arise. Additionally, the process of information spreading will start earlier as individual diverse behavior enhances. When the network_size increases gradually, the process of information propagation is delayed and the deviation of information propagation is increased, as shown in Figure 3b,d,f. Finally, our theoretical analyses (lines) match the outcomes of the simulation (symbols).

The combined impact of the parameters plane (λ, *a*) ((λ, *b*)) on $R\left(\infty \right)$ on the multi-layer ER network is examined in Figure 4. A crossover phenomenon appears as the value λ of increases. The diagram can then be split into two sections. The continuous phase transition with a second order is visible from region I as the increase in λ is in the growing pattern of the individual final spreading scale $R\left(\infty \right)$. As λ increases in region II, the growing $R\left(\infty \right)$ pattern displays a first-order discontinuous phase transition. The information spreading and the transition from a second-order continuous phase transition to a first-order discontinuous phase transition are both changed by the intensity of diverse behaviors. When network_size is reduced to 1000, a crossover phenomenon appears earlier. When network_size is increased to 15,000, a larger value of a(b) is required to produce the crossover phenomenon. This shows that information propagation is also influenced by the network_size.

### 5.2. Propagation Phenomena on a Two-Layer SF Network

Figure 5 shows the impact of the unit spreading possibility λ and diverse behavioral parameters *a* and *b* on the final adoption scale for the multi-layer SF network. Figure 5a,c demonstrate that, as λ increases, the final spreading scale $R\left(\infty \right)$ increases until it reaches global adoption. In subgraph (c) (v=4), the ultimate spreading size’s growing pattern shows a continuous phase transition with a second order when a=b=0.2. However, when a=b=0.5 and a=b=0.8, the growth of $R\left(\infty \right)$ reveals a discontinuous phase transition with the first order. Different from subgraph (Figure 5c), subgraph (Figure 5a) (v=2) shows a first-order discontinuous phase transition whatever the value of *a* and *b* are. Thus, when comparing subgraphs (Figure 5a) and (Figure 5c), it can be seen that the degree distribution heterogeneity can affect the increase pattern of the final spreading scale. The larger degree distribution heterogeneity can promote the global adoption of information.

The relative variances and crucial information spreading possibilities of Figure 5a,c, are shown in Figure 5b,d, respectively. The deviation of information spreading, which is represented by the highest values of relative variance χ, is where the global adoption information arises. Additionally, when the degree distribution heterogeneity rises, the process of information propagation will begin earlier. As a result, the conclusions of our theoretical analysis (lines) are consistent with those of the simulation (symbols).

Figure 6 illustrates $R\left(\infty \right)$ on the propagation parameter plane (λ, *a*)((λ,*b*)) for the multi-layer SF network. Then, using the parameters v=2 and v=4, respectively, the subgraphs (Figure 6a,b) demonstrate the growth tendency of $R\left(\infty \right)$. When the degree distribution heterogeneity is relatively small, i.e., v=4 in subgraph (Figure 6b), the pattern of the individual final spreading scale $R\left(\infty \right)$ changes from growing continuously in the phase transition with the second order in region I to growing discontinuously in the first-order phase transition in region II. Consequently, the diagram is split into two sections. Since there are active adoption behaviors in region I, the continuous phase transition with the second order can be seen in the change pattern of $R\left(\infty \right)$. The expanding pattern of $R\left(\infty \right)$ indicates the discontinuous phase transition with the first order, which is caused by the individuals’ increasingly negative behaviors in region II. The pattern of the individual ultimate spreading size $R\left(\infty \right)$, on the other hand, grows discontinuously in the first-order phase transition in the entire region when the degree distribution heterogeneity is substantially significant, as in subgraph (a), where v=2. Therefore, the heterogenous degree distribution can affect the growth of $R\left(\infty \right)$ and the information propagation.

## 6. Conclusions

The impact of the behavioral adoption diversity on the information propagation in the multi-layer networks is discussed in this study. We found that the same individual shows different or even opposite behaviors on different network layers. The impact of both active and negative LTI behaviors on information propagation is then separately considered on the multi-layer networks. Therefore, we offer an information propagation model, which includes two LTI behavioral features and a two-layer network. Meanwhile, we present an enhanced edge division theory to research the information spreading mechanism on the multi-layer networks. We discover various crossover events on information propagation using theoretical derivation and numerical simulation. The active LTI behaviors can encourage the information breakout on the multi-layer ER network. Moreover, a second-order continuous phase transition may be seen in the final spreading scale as the transmission probability rises. The final spreading scale displays a first-order discontinuous phase transition with the rise in the transmission probability when the individual behaves negatively. Furthermore, on the multi-layer SF network, the degree distribution heterogeneity can affect the propagation phenomena. The final spreading scale shows a first-order discontinuous phase transition with an increase in the transmission probability whenever the SF network has a strong degree distribution heterogeneity. The propagation pattern switches from second-order continuous phase transition to first-order discontinuous phase transition with the change in the LTI behavior intensity, but this is when the SF network has a weak degree distribution heterogeneity.

Information propagation relies heavily on behavior diversity on multi-layer networks, yet there is a dearth of comprehensive theoretical modeling and research in this area. We propose models and conduct qualitative and quantitative analyses of the network’s response to multiple behavior heterogeneity. The information propagation mechanism of a fresh scene is shown by our investigation. This paper focuses on the impact of adoption thresholds on individual adoption behavior on multi-layer networks, and can effectively draw corresponding conclusions. However, the influence of parameters such as individual activity heterogeneity and limited contact ability of multi-layer networks on the individual adoption ability was not involved in the discussion; this will be the direction of continued research in the future. In addition, whether the number of nodes in a complex network affects the results and whether an excessive number of nodes affects the speed of information propagation are the limitations of this study. 

## Figures and Tables

**Figure 1 entropy-25-00458-f001:**
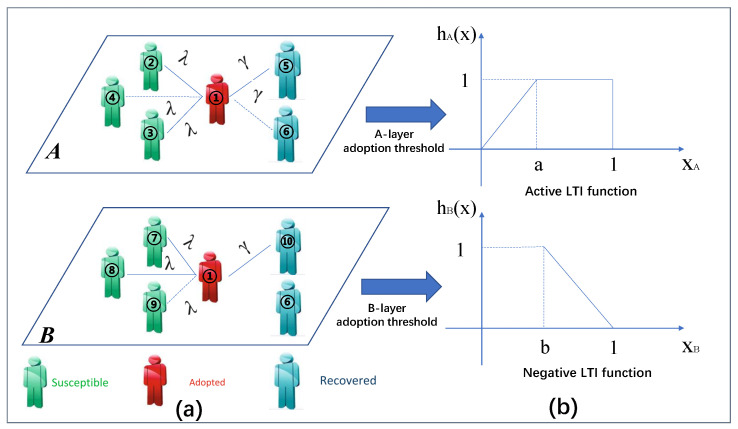
(**a**) **Diagram of information propagation on a two-layer network.** Individual 1 in A-state at time *t* will spread information. The transmission probability of the A-layer or B-layer is denoted by the symbol λ. The dashed lines represent links between individuals 1 and 4 in the A-layer and the individuals 1 and 9 in B-layer, thus indicating that the information cannot be spread. The explanation is that, prior to time *t*, the information has successfully been conveyed from individual 1 to individual 4 (or 9) via linkages in either the A-layer or B-layer. Solid lines indicate that each edge has not yet transmitted information. Furthermore, the structures of the two network layers are different, but the same adopted state individual 1 can exist in both network layers. The susceptible individuals around individual 1 are not the same as the recovered ones, which is similar to how individuals can publish speech messages on the Weibo platform or forward messages in the WeChat circle of friends. Individual 1 and individual 6 are adjacent in the A-layer network structure, but not in the B-layer network, so, even if the same individuals exist in different network layers, they cannot transmit information due to the different network layer structures, illustrating edge heterogeneity on multilayer networks. (**b**) **Probability model of diverse behaviors on a two-layer network.** The symbols $h_{A}\left(x\right)$ and $h_{B}\left(x\right)$ represent the proportion of the information obtained by an individual in the S-state in the A-layer and the B-layer to its degree, respectively. The symbols *a* and *b* represent the LTI parameters of the individual, respectively. $X_{A}$ represents the individual’s LTI parameter on the A-layer. $X_{B}$ represents the individual’s LTI parameter on the B-layer.

**Figure 2 entropy-25-00458-f002:**
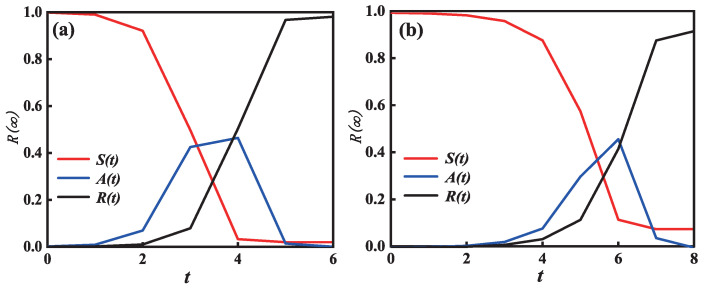
**Temporal variation in node density in different states.** In subgraph (**a**), a=b=0.1, it requires six steps to reach the maximum propagation scale; in subgraph (**b**), a=b=0.3, the propagation process requires eight steps; the proportion of susceptible individuals gradually decreases with time, the proportion of recovery individuals gradually increases with time, and the proportion of adopted individuals first increases and decreases to 0. The other basic parameter is λ = 0.3.

**Figure 3 entropy-25-00458-f003:**
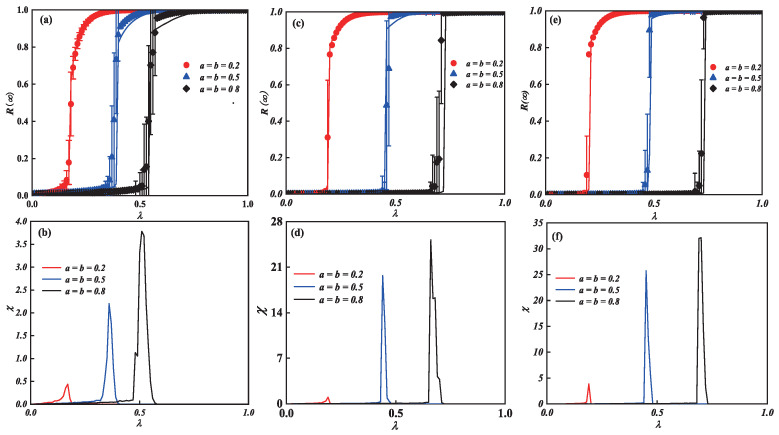
**Illustration of adoption behavioral diversity on multi-layer networks with unit transmission probability.** Subgraph (**a**,**c**,**e**) demonstrates how the parameters of behavioral diversity affect phase transition. The relative variances and the crucial values of (**a**,**c**,**e**) are shown in subgraph (**b**,**d**,**f**), respectively. Additionally, in subgraph (**a**,**c**,**e**), the solid lines on the diagram represent the theoretical anticipated results, whereas the symbols of the drawing denote the simulated results, where the network_size of the subgraph (**a**,**b**) is 1000, the network_size of subgraph (**c**,**d**) is 10,000, and the network_size of subgraph (**e**,**f**) is 15,000.

**Figure 4 entropy-25-00458-f004:**

**The combined influence of the diverse behavioral parameters a and b on each individual’s eventual spreading size for a multi-layer ER network.** In Figure 4, region I shows the continuous phase transition with a second order by final size R(∞) growing in the pattern. The discontinuous phase transition with the first order is visible from region II as a rising pattern of ultimate spreading scope R(∞), where the network_size of the subgraphs (**a**–**c**) are 1000, 10,000, and 15,000, respectively.

**Figure 5 entropy-25-00458-f005:**
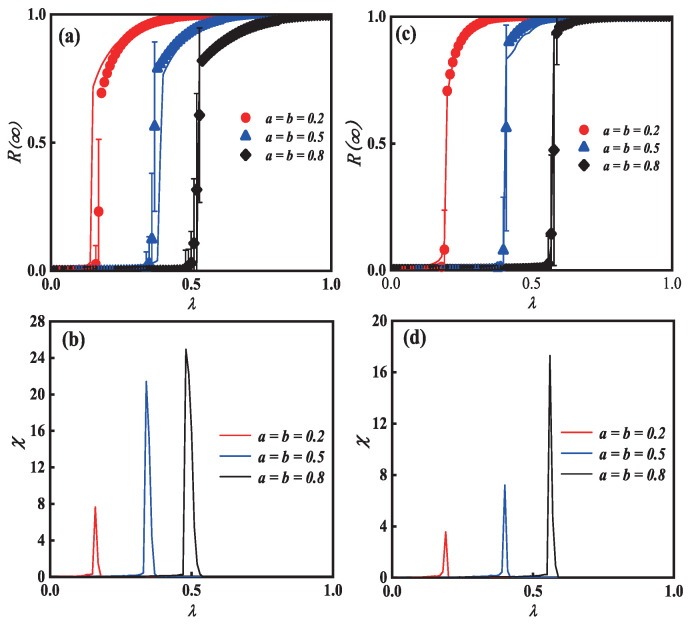
**The impact of diverse behavior parameters *a* and *b* on final spreading scale on the multi-layer SF network with unit transmission probability.** The horizontal subgraphs use the same degree distribution exponent, v=2 with subgraphs (**a**,**b**), meanwhile v=4 with subgraphs (**c**,**d**). The impact of on the final spreading scale with unit propagation probability λ are presented in subgraphs (**a**,**c**).

**Figure 6 entropy-25-00458-f006:**
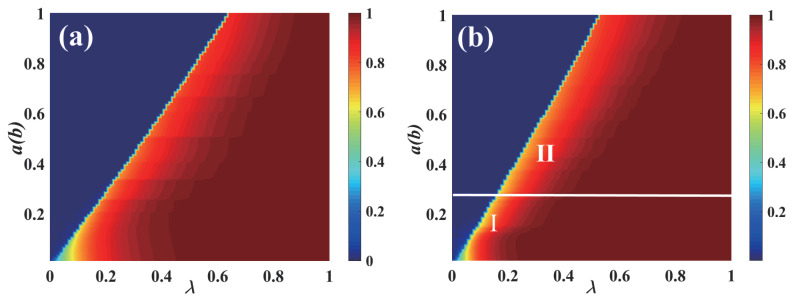
**The combined impact of the diverse behavior parameter *a* and *b* on the final spreading scale for the multi-layered SF network.** The impacts of (λ, *a*) and (λ, *b*) on the final spreading scale are shown in subgraphs (**a**,**b**) with v=2 and v=4, respectively. In subgraph (**a**), there is no crossover phenomenon of propagation. In subgraph (**b**), there is a crossover phenomenon of propagation.

## Data Availability

Not applicable.

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
