# Peer review of "Analysis of Influence of Behavioral Adoption Threshold Diversity on Multi-Layer Network"

_entropy, 2023, doi:10.3390/e25030458_

Round 1

Reviewer 1 Report

The paper by G. Deng et al. introduces a behavior adoption model on multi-layer networks. In my opinion, the paper is interesting. However, I suggest the authors take into account the issues below when revising the manuscript.

(1) Of greatest concern is how the model responds to real-world examples. Could the authors provide some straight-forward examples of real world scenarios where information propagates in two-layer networks like the example in Fig. 1?

(2) The evidence for the “first-order phase transition” is very weak. More solid evidence is required.

(3) The literature review is a little weak. More research related to the multi-layer network models in statistical physics and complex networks needs to be reviewed. 

(4) The conclusion part is too brief. The authors can extend the discussion to possible future extensions of their current work. A discussion of the limitations of the work can also be added.

(5) There are some grammatical errors in the paper. I suggest that authors carefully proofread their manuscript before submitting a revised manuscript.

Reviewer 2 Report

In this paper, the authors studied the spreading of behavior in double layered networks with heterogeneous adoption thresholds. The adoption probability of nodes is non-uniform, and in particular, a specific pattern of adoption probability is proposed. Mean-field equations of the model were solved and numerical calculations were performed. As a result, they found different phase transitions depending on individuals’ thresholds.

However, this paper is a small modification of the previously models, and the results are not interesting compared to the previous results of the existing models. In addition, the authors did not cite appropriate references, i.e., the papers that proposed models were excluded. Moreover, there is no rationale or justification for introducing the specific form of adoption probability. Why should there be such a form of adoption probability? Are there special results that cannot be seen in other forms?

Aside from the content, English grammar also needs to be improved. In addition, qchat or weibo is used as an example in the text, but the basic introduction of what they are should be explained to those who do not know it.

As a result, the evidence of the introduced model and conclusion is thin. In addition, the reported results are not a major step forward compared to the existing results or models, so it is insufficient to be published in Entropy.

Reviewer 3 Report

The authors used the SAR diffusion model on the multi-layered network to show the effect of behavioral adoption diversity on the propagation of information. This paper is not very creative or surprising, but it seems that it has a meaningful contribution. Therefore, I believe it could be considered published in Entropy after revising the following suggestions.

-       The model explanation in Section 2 is not well written, and it is hard to understand the model. There is no definition of h_A and h_B. Is h_A a function of ‘x’ or a function of ‘x and a’? It is not consistent: h(x) in figure 1 and h(x, a) in equations (1) and (2). What is x?

-       The authors suggested two equations of h in equations (1) and (2), which have different shapes of functional forms. Why do authors choose these functions? How h_A and h_B are derived? What if h varies? How this variation affects the results, such as the type of phase transition?

-       The network structure in the two layers are identical? Authors might be able to run the simulations on the networks with negatively/positively/neutrally correlated layers.

-       The node size in this paper is 10,000. To see the critical behavior near the phase transition, finite-size scaling might be useful. Are there any needs to observe the simulation with different sizes to see the type of phase transition more accurately? Generally, mathematical results coincide with the simulation results with thermodynamic limits; that is, the system size is infinity. Therefore, finite-size scaling might be needed to compare the mathematical analysis to the numerical one.

Round 2

Reviewer 1 Report

The authors have addressed almost all the comments I raised in my review report. However, I am still not satisfied with the first-order transition clarification performed by the authors. In order to be sure of the issue, at least a method such as finite-size scaling or analytic approach should be performed. Therefore, I cannot recommend the paper in the current version, and I suggest that the authors revise the manuscript significantly to publish.

Reviewer 2 Report

The comments that previously raised have been appropriately answered.

Author Response

Thanks for the suggestions.

Reviewer 3 Report

It does not seem that the authors addressed the reviewers’ comments well. Therefore, I do not recommend publishing this paper in this journal with the current version. Authors should revise the manuscript significantly to publish it.

·      As Reviewer 2 pointed out, the evidence for the “first-order phase transition” is very weak. The authors just added the simulation with size 1000. However, as I mentioned in the first round review, “finite-size scaling (FSS)” might be needed to show first-order phase transition. Authors can find many literatures on FSS from google scholar and so on.

·      What are ER and SF? Are they abbreviations of Erdos-Rényi and Scale-free? I could not find any explanation for this.

·      Are two layers the same structure? Node 2 and node 7 in figure 1 are identical? Nodes 1 and 6 are connected in the first layer, but they are not connected in the second layer in figure 1. Therefore, it does not seem that the structure of the two layers is identical. If two layers are not identical, as I commented in the first round review, authors should clarify the relation of the network structure of two layers: positively/negatively/neutrally correlated multi-layered network. If two layers are identical, is it really a TWO-layered network? It is hard to find the reason why authors should use the concept of a multi-layered network. According to the explanation in lines 117-131, two layers should not be identical.

·      What is the rationale for two transition rates, h_A and h_B? The motivation and justification for why authors should consider multiple transition rates are not sufficiently explained.

·      Why/how did the focal node choose one layer among multiple layers to spread information to its neighbors? Is it randomly chosen when information spreads or assigned beforehand when network ties are made?

Round 3

Reviewer 1 Report

The authors addressed the comments raised in the previous review round. I now recommend the paper for publication.